# Comprehensive Analysis of Virulence Determinants and Genomic Islands of *bla*_NDM-1_-Producing *Enterobacter hormaechei* Clinical Isolates from Greece

**DOI:** 10.3390/antibiotics12101549

**Published:** 2023-10-18

**Authors:** Angeliki Mavroidi, Konstantina Gartzonika, Nick Spanakis, Elisavet Froukala, Christos Kittas, Georgia Vrioni, Athanasios Tsakris

**Affiliations:** 1Department of Microbiology, General University Hospital of Patras, 26504 Patras, Greece; amavroidi@live.com; 2Department of Microbiology, Medical School, Ioannina University Hospital, 45110 Ioannina, Greece; kgartzon@uoi.gr (K.G.); ckittas@gmail.com (C.K.); 3Department of Microbiology, Medical School, University of Athens, 11527 Athens, Greece; nespanakis@gmail.com (N.S.); elisavetfrou@gmail.com (E.F.); gvrioni@med.uoa.gr (G.V.)

**Keywords:** *Enterobacter cloacae* complex, whole genome sequencing, genomic islands, multidrug-resistance, virulence factors, mobile genetic elements

## Abstract

Nosocomial outbreaks of multidrug-resistant (MDR) *Enterobacter cloacae* complex (ECC) are often reported worldwide, mostly associated with a small number of multilocus-sequence types of *E. hormaechei* and *E. cloacae* strains. In Europe, the largest clonal outbreak of *bla*_NDM-1_-producing ECC has been recently reported, involving an ST182 *E. hormaechei* strain in a Greek teaching hospital. In the current study, we aimed to further investigate the genetic make-up of two representative outbreak isolates. Comparative genomics of whole genome sequences (WGS) was performed, including whole genome-based taxonomic analysis and in silico prediction of virulence determinants of the bacterial cell surface, plasmids, antibiotic resistance genes and virulence factors present on genomic islands. The enterobacterial common antigen and the colanic antigen of the cell surface were identified in both isolates, being similar to the gene clusters of the *E. hormaechei* ATCC 49162 and *E. cloacae* ATCC 13047 type strains, whereas the two strains possessed different gene clusters encoding lipopolysaccharide O-antigens. Other virulence factors of the bacterial cell surface, such as flagella, fimbriae and pili, were also predicted to be encoded by gene clusters similar to those found in *Enterobacter* spp. and other Enterobacterales. Secretion systems and toxin–antitoxin systems, which also contribute to pathogenicity, were identified. Both isolates harboured resistance genes to multiple antimicrobial classes, including β-lactams, aminoglycosides, quinolones, chloramphenicol, trimethoprim, sulfonamides and fosfomycin; they carried *bla*_TEM-1_, *bla*_OXA-1_, *bla*_NDM-1_, and one of them also carried *bla*_CTXM-14_, *bla*_CTXM-15_ and *bla*_LAP-2_ plasmidic alleles. Our comprehensive analysis of the WGS assemblies revealed that *bla*_NDM-1_-producing outbreak isolates possess components of the bacterial cell surface as well as genomic islands, harbouring resistance genes to several antimicrobial classes and various virulence factors. Differences in the plasmids carrying β-lactamase genes between the two strains have also shown diverse modes of acquisition and an ongoing evolution of these mobile elements.

## 1. Introduction

*Enterobacter hormaechei* and *E. cloacae* are members of the *Enterobacter cloacae* complex (ECC), frequently recognised as the causative agents of nosocomial infections, including pneumonia, urinary tract and soft-tissue infections, septicaemia, and meningitis [1]. The species/subspecies assignment of the genus *Enterobacter* has often been inconsistent via routine identification techniques, 16S *rRNA* and housekeeping genes, while the reclassification of the species/subspecies via phylogenetic studies based on whole genome DNA-DNA hybridisations and sequencing is ongoing [2,3,4,5]. Genome-wide analysis of ECC type strains, including both clinical and environmental isolates, has revealed the presence of highly conserved genes associated with the survival of ECC strains, whereas genes involved in drug resistance and virulence were present in different genomic regions [6]. The surface of the bacterial cell is composed of common components, which contribute to the cell wall rigidity and the interaction of the bacteria with the environment, host immune response and host–pathogen interactions [7,8].

ECC are members of the ESKAPE pathogens, exhibiting resistance to several, if not all, antimicrobial classes, and are therefore described as multidrug-resistant (MDR) pathogens [9,10]; https://arpsp.cdc.gov/profile/antibiotic-resistance/mdr-enterobacter (last accessed on 1 August 2023). The most common antimicrobial resistance mechanisms in such microorganisms include the acquisition of mobile genetic elements (MGEs) from other Enterobacterales and the over-expression of efflux pumps [1]. The predominant cause of resistance to β-lactams is the production of β-lactamases, which inactivate the drug by hydrolysing the β-lactam ring. Based on the Ambler scheme, β-lactamases are grouped into four classes: the class B metallo-enzymes, also known as metallo-β-lactamases (MBLs); and the serine-reactive hydrolases (classes A, C and D) [11]. ECC strains are intrinsically resistant to cephalosporins due to the production of the chromosomal cephalosporinase *bla*_ACT-16_, whereas the spread of acquired β-lactamases has been often documented, such as the class A extended-spectrum beta-lactamases (ESBLs) of the CTX-M and SHV families as well as carbapenemases, such as the VIM- and NDM-types (class B), KPC-2 (class A) and OXA-48 (class D) [1,7,8,12,13].

Nosocomial outbreaks of MDR ECC are associated with a small number of MLST STs, such as ST78, ST114 and ST171, compared with the overall diversity of the STs identified in *Enterobacter* spp. [12,13,14]. Similarly, with other Enterobacterales, there were no obvious associations of NDM producers and STs or well-defined *bla*_NDM_-producing international clones [14]. ST182 has been sporadically reported in several countries, while it was associated with outbreaks in the Czech Republic and Greece [15,16,17]. In the Czech Republic, the first *bla*_NDM-4_-producing ST182 isolate was reported in 2012 and a clonal expansion of these isolates was observed during 2016 [15,16]. The largest outbreak in Europe of *bla*_NDM-1_-producing ST182 isolates has been recently reported in a retrospective study of carbapenem-non-susceptible ECC, during a 6-year period (March 2016–March 2022) in the University Hospital of Ioannina, Greece [17].

In the present study, we aimed to further investigate the genetic make-up of *bla*_NDM-1_-producing ECC strains from the aforementioned outbreak in Greece. For this purpose, we have compared the whole genome sequences of two *bla*_NDM-1_-producing ECC isolates, a representative of ST182 strain EC-ML559 and strain EC-ML621, which co-produced the ESBL *bla*_CTX-M-15_ and belonged to a closely related pulsed-field gel electrophoresis (PFGE) subtype to the major subtype of the clonal outbreak. In silico characterisation and comparative genomics were performed for the gene clusters encoding the major virulence components of the cell surface of Gram-negative bacteria, such as the enterobacterial common antigen (ECA), colanic acid (CA), and O-antigen lipopolysaccharide (LPS), pili, flagella and fimbriae [7,8]. Additionally, we have explored the presence of genomic islands, which carried acquired plasmids, antimicrobial resistance and virulence genes, including secretion systems and toxin–antitoxin systems [18,19,20].

## 2. Results

### 2.1. Antimicrobial Susceptibility Testing, MLST, WGS and Type (Strain) Genome Server (TYGS) Analysis

The antibiotic susceptibility testing of strains EC-ML559 and EC-ML621 has revealed that both strains were multidrug-resistant, e.g., resistant to β-lactams (including carbapenems), β-lactam/β-lactamase inhibitor combinations (amoxicillin/clavulanic, ticarcillin/clavulanic acid, piperacillin/tazobactam, acid), quinolones (ciprofloxacin, ofloxacin), trimethoprim/sulfamethoxazole, tobramycin (Appendix A). The characteristics of the WGS assemblies, MLST allelic profiles, predicted genomic islands and MGEs of strains EC-ML559 and EC-ML621 are presented in Table 1. Strain EC-ML621 was assigned to the novel MLST ST2143 (available at https://pubmlst.org/organisms/enterobacter-cloacae, last accessed on 1 August 2023), differing only in one nucleotide position in the MLST alleles (*dna*A-554) compared with strain EC-ML559 of ST182 (*dna*A-49), as shown via MLST and WGS. The type-based species and subspecies clustering of 31 *Enterobacter* spp. strains yielded 25 species clusters (average branch support of 92.5%) and 26 subspecies clusters (Appendix A). The provided query strains EC-ML559 and EC-ML621 belonged to the same cluster with the *E. hormaechei* type strains, being closely related to the *E. hormaechei* subsp. *xiangfangensis* LMG27195 type strain; thus, they were assigned as *E. hormaechei*.

### 2.2. Enterobacterial Common Antigen (ECA), Colanic Acid (CA) and Lipopolysaccharide (LPS) O-Antigen

In strains EC-ML559 and EC-ML621, gene clusters encoding proteins involved in the biosynthesis of the enterobacterial common antigen (ECA), colanic acid (CA) and lipopolysaccharide (LPS) O-antigens were identified (Figure 1). The ECA cluster, known as the *wec* operon from *wecA* to *wecG*, of strains EC-ML559 and EC-ML621 was more similar to *E. hormaechei* ATCC49162 than *E. cloacae* ATCC 13047. The ECA is a carbohydrate antigen built of repeating units of three amino sugars, the structure of which is conserved throughout Enterobacterales [21]. The ECA cluster encodes proteins involved in many steps of ECA synthesis, such as the addition of ECA polysaccharide chains to a lipid carrier and the transportation of ECA to the bacterial cell surface [21]. The CA gene cluster, within the *wza* and *galF* genes, includes genes for the production of the extracellular polysaccharide CA or M antigen, and synthesis of GDP-L-fucose, one of the precursors of CA, which is widely found within *E. coli*, as well as within other species of the Enterobacterales [22]. The CA cluster of strains EC-ML559 and EC-ML621 was highly similar to the CA cluster of the *E. cloacae* ATCC13047 type strain, whereas no CA cluster was identified in the *E. hormaechei* ATCC49162 type strain.

The (LPS) O-antigen biosynthetic loci were also located on gene clusters within the housekeeping genes *galF* and *gndA* (Figure 2), as described in *E. coli* and *E. cloacae* isolates [23,24]. Phylogenetic analysis of the O-antigen clusters of strains EC-ML559 and EC-ML621 was performed at the NCBI database using other ECC type strains, which revealed correlations of the O-antigen clusters with the subspecies level of the type strains (Appendix A). The O-antigen biosynthetic locus of strain EC-ML559 was highly similar to the *E. hormaechei* strain CUVET18-121 (GenBank: CP114979.1; region: 3040072–3053366; 99.99% identity) from Thailand, the *E. hormaechei* strain Ehh_23 (GenBank:CP126767.1; region: 2958890–2972184; 98.98% identity) and the *E. hormaechei* strain Ehh_24 (GenBank: CP126761.1; region: 2960240–2973534; 98.98% identity) from Switzerland, and 70% identical to the *E*. *coli* strain RHB04-C20 (GenBank CP055786; region: 1810262–1823551). All these clusters contained the *rfbB, -D, -A, -C* (or *rmlB*, *-D, -A, -C*) genes, which encode proteins for dTDP-rhamnose synthesis. Analysis of the WGS assemblies in the *E. cloacae* PubMLST database (avalaiable at https://pubmlst.org/organisms/enterobacter-cloacae, last accessed on 1 August 2023) has shown that the *E. hormaechei* strain CUVET18-121 belonged to ST182, whereas strains *E. hormaechei* Ehh_23 and *E. hormaechei* Ehh_24 were of a different ST (ST245). Notably, the O-antigen biosynthetic locus of strain EC-ML621 had a different genetic structure, being highly similar to the locus of an *E. hormaechei* strain Eh202_LBHALD (GenBank CP129250, region: 507656–523007) from Senegal, which belonged to ST182. Moreover, by using the Mobile Element Finder tool, it was revealed that the *E. hormaechei* strains CUVET18-121, Ehh_23, Ehh_24 and Eh202_LBHALD carried only the chromosomal *bla*_ACT-16_.

### 2.3. Flagella, Fimbriae and Secretion Systems

Strains EC-ML559 and EC-ML621 possessed *flag*-3a loci, which encode peritrichous flagellar systems, as described previously in the *E. cloacae* ATCC 13047 and other *Enterobacter* spp. isolates [25,26]. The *flag*-3a loci all comprised three gene blocks occurring in the following order: block 1, *flhEAB-cheZYBR-fliEFGHIJKLMNOPQR*; block 2, *fliZACDST;* and block 3, *flgNMABCDEFGHIJKL-flhDC-motAB-cheAW*. Both strains harboured the *fimA-Z* gene clusters encoding type I fimbriae (or type I pili) biosynthesis proteins [27,28,29], which were highly similar (>99.9%) to the *E. hormaechei* subsp. *xiangfangensis* strain Eh8322_LBHALD (GenBank CP129636.1, region: 1872180–1878869) from Senegal, *E. hormaechei* strain CUVET18-121 (CP114979, region: 1131840–1138529) from Thailand and other *E. hormaechei* strains. Moreover, both strains also possessed a second fimbriae gene cluster, which was also identified in the aforementioned strains. Notably, both fimbriae gene clusters were absent from the *E. hormaechei* ATCC 49162 type strain.

BlastN comparisons showed that the T2SS, T6SS and T4P1 secretion systems of strains EC-ML559 and EC-ML621 were more similar to the respective gene clusters of the *E. hormaechei* ATCC 49162 type strain than the *E. cloacae* 13047 type strain (Figure 3). Both strains possessed T4P1 gene clusters encoding pili biosynthesis proteins [30,31], containing the *hofQ-hof-hofO-pilN-pilM* genes (GenBank, JASKGQ010000009.1; region 43949–45669). Strain EC-ML621 carried two more pili biosynthesis loci; the first locus (contig JASKGQ010000016.1, region 1: 77466–86705) was similar to the *E. hormaechei* strain 21KM1498 plasmid pKM98_p3 (GenBank CP126866.1) and *E. ludwigii*_strain CM-TZ4 (GenBank CP116349.1), while the second locus (JASKGQ010000022, region: 20641–30420) was located on the *bla*_LAP2_-carrying plasmid sequence, which was similar to the *E. hormaechei* subsp. *xiangfangensis* strain 090095 pLAP2_090095 (GenBank CP128418.1).

### 2.4. In Silico Prediction of Genomic Islands

In the identified genomic island (Appendix A), both EC-ML559 and EC-ML621 isolates harboured β-lactamase-carrying contigs highly similar to published plasmid sequences (Appendix A), including the *E. cloacae* pECL_A plasmid carrying *bla*_TEM-1_ [3]. Both isolates carried the *bla*_OXA-1_ gene, but it was not possible to determine in silico whether it was located on a plasmid or on the chromosome due to the small sizes of the obtained *bla*_OXA-1_-harbouring contigs and the abundance of the presence of *bla*_OXA-1_ genes among different plasmid types and bacterial species. The *bla*_NDM-1_-harbouring contigs of both isolates were highly similar to DNA fragments of *Klebsiella pneumoniae* subsp. *pneumoniae* strain KPX plasmid pKPX-1 (GeneBank: AP012055.1) and *E. hormaechei* subsp. *xiangfangensis* strain ST114 plasmid pLAU_ENM30_NDM1 (GeneBank: MN792917.1) [32,33] Of note, the genetic background of *bla*NDM-1 differed in strains EC-ML559 and EC-ML621 (Appendix A). Moreover, strain EC-ML621 also possessed *bla*_CTX-M-14_-, *bla*_CTX-M-15_- and *bla*_LAP-2_-harbouring plasmid sequences (Table 1, Appendix A). The *bla*_CTX-M-14_ gene was carried on a contig (JASKGQ010000024.1), which also possessed an IncL/M genetic element, showing 99.6% similarity to the *repC*, *repB* and the replication initiation protein-encoding gene *repA* of the *K. pneumoniae* plasmid pMU407.1 (GenBank: U27345.1). The *bla*_LAP-2_ genetic background was highly similar to the *E. hormaechei* subsp. *xiangfangensis* strain 090095 plasmid pLAP2_090095, GenBank Accession no. CP128418, WGS: JAHEVO010000000) and other Enterobacterales [34,35,36]. The latter strain belonged to a different MLST ST (ST175) and shared 33 out of 53 rMLST loci compared to strain EC-ML621, as shown by analysing the WGS assembly in the *E. cloacae* PubMLST database (avalaiable at https://pubmlst.org/organisms/enterobacter-cloacae, last accessed on 1 August 2023).

In the predicted genomic islands, besides the presence of β-lactamase genes, both strains EC-ML559 and EC-ML621 carried resistance genes to several antimicrobial classes (e.g., aminoglycosides, chloramphenicol, quinolones, fosfomycin, trimethoprim, sulfonamides, macrolides, quaternary ammonium compounds), heavy-metal resistance genes to arsenic copper/silver and tellurium and genes associated with bacterial virulence, including type II and type IV toxin–antitoxin systems, secretion systems (T1SS, T2SS, T4SS, T6SS and T4P), flagellin and fimbriae genes (Table 1). The origin of the transfer site (*oriT*), the relaxase gene, the gene encoding type IV coupling protein (T4CP) and the gene clusters for bacterial type IV secretion system (T4SS) identified in strains EC-ML559 and EC-ML621 are presented in Table 2.

## 3. Materials and Methods

### 3.1. Bacterial Isolates, Antimicrobial Susceptibility Testing, Whole Genome Sequencing (WGS) and MLST Typing

Susceptibility testing of strains EC-ML559 and EC-ML621 to various antimicrobial agents has been performed using the automated system MicroScan (Beckman Coulter, Brea, CA, USA), according to the manufacturer’s instructions. The categorisation of the isolates and interpretation of the minimum inhibition concentrations (MICs) of the antimicrobial agents were based on the European Committee on Antimicrobial Susceptibility Testing (EUCAST) criteria and breakpoints for Enterobacterales (available at: https://www.eucast.org/fileadmin/src/media/PDFs/EUCAST_files/Breakpoint_tables/v_13.0_Breakpoint_Tables.pdf, last accessed on 1 August 2023). The genome sequence of the representative ST182 outbreak strain EC-ML559 has been presented in a previous study [17]; (GenBank Accession no. JARUPS000000000.1). Strain EC-ML621 was cultured in LB medium and allowed to grow until the mid-logarithmic phase at 30 °C in a shaking incubator. DNA was extracted using a genomic DNA isolation kit (Promega, Madison, WI, USA). The concentration and quality of the DNA were determined via Nanodrop 2000 (Thermo Fisher Scientific, Dreieich, Germany). Multilocus-sequence typing (MLST) was performed via the amplification of the seven housekeeping genes, using primer sequences described in *E. cloacae* MLST database (PubMLST web site https://pubmlst.org/ecloacae, last accessed on 1 August 2023)). The isolate EC-ML621, the novel allele and ST identified were deposited in the PubMLST database under PubMLST id 1364 [37]; (https://pubmlst.org/bigsdb?page=info&db=pubmlst_ecloacae_isolates&id=1364, last accessed on 1 August 2023). Moreover, WGS of isolate EC-ML621 was performed on an Illumina platform (Eurofins Genomics Europe Sequencing GmbH, Konstanz, Germany). After sequence cleaning and normalisation, high-quality reads were assembled using SPAdes [38].

### 3.2. Whole-Genome-Based Taxonomic Analysis

Whole-genome-based taxonomic analysis of the ECC strains EC-ML559 and EC-ML621 was performed by using the free bioinformatics platform Type (Strain) Genome Server (TYGS) available at https://tygs.dsmz.de (last accessed on 23 July 2023) [39], while information on nomenclature, synonymy and associated taxonomic literature was provided by the List of Prokaryotic names with Standing in Nomenclature (LPSN, available at https://lpsn.dsmz.de; last accessed on 23 July 2023) [40]. Briefly, the TYGS analysis included the determination of closely related type strains, pairwise comparison of genome sequences and phylogenetic inference via the MASH algorithm. The resulting intergenomic distances were used to infer a balanced minimum evolution tree with branch support via FASTME 2.1.6.1, including SPR post processing. Branch support was inferred from 100 pseudo-bootstrap replicates each. The trees were rooted at the midpoint and visualised with PhyD3. The type-based species clustering using a 70% dDDH radius around each of the 28 type strains, and subspecies clustering was carried out using a 79% dDDH threshold.

### 3.3. In Silico Prediction of Genomic Islands, Mobile Elements, Antimicrobial Resistance and Virulence Genes

Genomic island prediction was performed for strains EC-ML559 and EC-ML621 by using the IslandViewer4 web application (available at: https://www.pathogenomics.sfu.ca/islandviewer; last accessed on 10 August 2023), which includes all genomic islands found on published sequenced bacterial and archaeal genomes that have been predicted using the currently most accurate GI prediction methods, including IslandPick, SIGI-HMM, IslandPath-DIMOB [41]. The *E. hormaechei* subsp. *xiangfangensis* LMG27195 type strain genome sequence was used as a reference sequence for the alignment of each genome sequence. The presence of mobile elements, antimicrobial resistance and virulence genes was predicted by using the Mobile Element Finder, Plasmid Finder. ResFinder and Virulence Finder tools available at: https://www.genomicepidemiology.org/services/ (last accessed on 10 August 2023), Center for Genomic Epidemiology, Technical University of Denmark, Denmark [42].

The Artemis Comparison Tool version 18.2.0 [43] and SnapGene version 5.1.5 (GSL Biotech LLC., San Diego, CA, USA) were used to manipulate and visualise the sequences studied. The oriTfinder tool [44] was used to explore the presence of conjugative regions of the self-transmissible MGEs: the origin of transfer site (*oriT*), the relaxase gene, the gene encoding type IV coupling protein (T4CP) and the gene cluster for bacterial type IV secretion system (T4SS). BlastN comparisons of the beta-lactamase-carrying contigs of the isolates with published plasmid sequences were performed by using the BLAST Ring Image Generator (BRIG) version 0.95 software (available at: https://brig.sourceforge.net/ (last accessed on 23 August 2023)) [45]. BlastN comparisons and visualisations of the gene clusters encoding ECA, CA, O-antigens and secretion systems of the ECC strains EC-ML559 and EC-ML621 were performed by using the Easyfig version 2.2.5 software [46], (available at: http://mjsull.github.io/Easyfig/ (last accessed on 23 August 2023)).

## 4. Discussion

The genus *Enterobacter* comprises species/subspecies often recognised as human, animal, and plant pathogens [1]. The difficulties in the species/subspecies assignment of the genus *Enterobacter* spp. and the definition of international clones among the diverse population structure of ECC strains are challenging for the diagnosis of infections caused by these pathogens and outbreak analysis [2,3,4,5]. Nosocomial infections caused mainly by MDR *E. cloacae* and *E. hormaechei* are frequently reported and are difficult to treat. A recent investigation of an outbreak in new-borns with *E. cloacae* complex isolates revealed, via WGS, *E. bugandensis* as the causative agent, and highlighted the need for better discrimination of *Enterobacter* species inside the ECC [47]. Moreover, the arsenal of these pathogens to acquire antimicrobial resistance determinants and virulence factors via mobile genetic elements from other Gram-negative bacteria further complicates the diagnosis and treatment of infections caused by ECC. In the present study, we have performed a comprehensive whole genome analysis of two representative *E. hormaechei bla*_NDM-1_-producing isolates (strains EC-ML559 and EC-ML621) recovered during a large outbreak in a Greek teaching hospital [17]. The isolates, which belonged to the closely related MLST STs 182 and 2143, were assigned as *E. hormaechei* via whole genome-based taxonomic analysis. In silico prediction and comparison of bacterial cell surface components and other systems involved in the pathogenesis of this microorganism were performed, including genomic islands, which often carry antimicrobial resistance genes and virulence factors, such as those involved in bacterial secretion, host determination and the colonisation of different strains.

Antigenic and phase variation mechanisms have been proposed to enable immune evasion during chronic or recurrent infection and/or the generation of variants with altered ability to colonise different niches in the host [8]. Variation in all major surface antigens is a hallmark of pathogens persistently colonising mucosal surfaces. Polysaccharides are important constituents of the bacterial cell surface and prominent antigens, including the enterobacterial common antigen, colanic acid or M-antigen and the lipopolysaccharide of Gram-negative bacteria [7,8]. Gene clusters involved in the biosynthesis of the ECA and CA were identified in both strains, EC-ML559 and EC-ML621, being similar to the respective gene clusters of the *E. hormaechei* ATCC 49162 and *E. cloacae* ATCC 13047 type strains. Gram-negative bacteria also produce LPS, which is composed of an O-antigen, an outer core, and an inner core all joined by covalent bonds [23,24]. The O-antigens are structurally and serologically diverse, allowing for escape from the immune system of the host. Additionally, antisera against O-antigens provide a basis for serotyping schemes in many Gram-negative bacteria and a basic tool utilised in outbreak investigations and epidemiological surveillance. The LPS O-antigens of *E. coli* have been well characterised [23], and, in a recent study, the molecular characterisation of *E. cloacae* O-antigens has been performed [24]. In the present study, the *E. hormaechei* strains EC-ML559 and EC-ML621 were found to possess different types of O-antigen clusters, although they belonged to closely related MLST STs, differing only in one MLST allele by one nucleotide difference. Moreover, the same O-antigen cluster of strain EC-ML559 was identified in *E. hormaechei* strains of different STs (ST182 and ST245), sharing similarities with a gene cluster found in an *E. coli* strain, suggesting the horizontal spread of these gene clusters via recombination and evolution in different strains of these species.

Among the components of the bacterial cell surface also associated with pathogenicity are filamentous organelles, such as flagella, pili and fimbriae [7,8]. These structures are involved in bacterial motility and adherence, biofilm formation and colonisation on infected hosts, acting as virulence factors. The bacterial flagellum consists of a filament core, which is preserved across bacterial species, while the outer domains exhibit high variability, and, in some cases, are even completely absent [25]. The Enterobacterales encode five distinct flagellar systems and the genes required for the assembly, maintenance and functioning of this flagellum are organised in five genomic clusters (*flag*-1 to *flag*-5) [26]. Both strains, EC-ML559 and EC-ML621, carried *flag*-3a peritrichous flagellar loci, which are restricted to members of three genera: *Enterobacter*, *Erwinia*, and *Pantoea*. All Gram-negative bacterial species that have been examined to date were found to produce one or more types of fimbriae with adhesive properties, which are involved in the specific adherence to receptors present on certain host cell types and the colonisation of the host [27,28,29]. In a previous study, at least 16 fimbrial gene clusters were identified in two enterohaemorrhagic *E. coli* O157:H7, and six of them were detected only in the *E. coli* O157 and O145 serotypes tested, while the *fimB-H* operon has been previously characterised in *E. coli* K12 strain and uropathogenic *E coli* [7,28,29]. Strains EC-ML559 and EC-ML621 also carried fimbrial genes, which were found in several *E. hormaechei* strains, which may contribute to pathogenesis. Type IV pili are also filamentous structures of the bacterial cell surface widespread among the β-, γ-, and δ-proteobacteria and the cyanobacteria, which contribute to motility and act as virulence factors [30,31]. In strain EC-ML621, two of the three pili biosynthesis gene clusters that were identified showed similarities with published plasmid sequences, suggesting that these clusters can be transferred horizontally between different strains or species.

Secretion system proteins aid in the transfer of a variety of virulence determinants across the bacterial cell membrane [18]. Because of the remarkable structural and functional similarities, secretion systems may be exploited for the development of novel antibacterial compounds. Recent studies have shown that secretion systems, including T6SS accessory and core components, T4SS, and multidrug resistance genes/efflux system genes seemed vital for the survival of bacteria in various environmental niches, such as humans and plants. The type IV secretion system (T4SS) is evolutionarily related to the conjugation system of bacteria and it was speculated that the low GC content and the presence of T4SS may be linked with horizontal gene transfer (HGT), which helps bacteria adapt to environmental changes and acquire antibiotic resistance. In a recent study of 952 *K. pneumoniae* strains, the presence of one conserved T2SS, one conserved T5SS, and two conserved T6SS was detected in more than 90% of the strains, whereas T1SS and T4SS were enriched in the hypervirulent and classical multidrug resistance pathotypes of *K. pneumoniae*, respectively [19]. In ECC, T2SS genes were absent in the *E. kobei, E. hormaechei*, and *E. ludwigii*, type strains previously studied, but there is a lack of experimental data to link the absence of T2SS with the pathogenicity of these strains. In the present study, the T1SS, T2SS, T4SS, T6SS and T4P secretion systems were identified in genomic islands present in both studied isolates, showing similarities with the *E. hormaechei* ATCC 49162 and *E. cloacae* ATCC 13047 type strains, whereas Type IV pili biosynthesis gene clusters (T4SS) were located on plasmid sequences of strain EC-ML621. Furthermore, both strains possessed toxin–antitoxin systems, which are abundant in bacterial genomes, in which they move via horizontal gene transfer. Specific toxin–antitoxin systems have been involved in the protection against phages, and may be involved in pathogenicity [20].

Finally, both isolates possessed plasmid sequences that carried antimicrobial resistance and virulence genes. In regard to β-lactamase genes, the *bla*_OXA-1,_
*bla*_TEM-1_-carrying *E. cloacae* pECLA-type and the *bla*_NDM-1_
*K. pneumoniae* pKPX-1-type plasmid sequences were found in both strains, whereas strain EC-ML621 also carried *bla*_CTX-M-14_-, *bla*_CTX-M-15_- and *bla*_LAP-2_-harbouring plasmid sequences. The *bla*_CTX-M-14_- and *bla*_CTX-M-15_-harbouring plasmids are ubiquitous in Enterobacterales. The Ambler class A LAP-1 β-lactamase was first characterised in 2007 [34], and shows a narrow hydrolysis spectrum against β-lactams inhibited by clavulanic acid. The *bla*_LAP-1_ gene was located on the same plasmid with the quinolone resistance gene *qnrS1*, which was identified in *E. cloacae* isolates from France and Vietnam. LAP-2 is a variant of LAP-1, which was first identified in an *E. cloacae* clinical isolate from China in 2008 [35]. *Enterobacter* spp. strains carrying the *bla*_LAP-1_ gene have been reported in Spain, Algeria and Korea, while *bla*_LAP-2_ has been detected in species of the genera *Klebsiella*, *Enterobacter* and *Salmonella* from China, Tunisia, Norway and the Netherlands [36].

The spread of antimicrobial resistance and virulence genes is often mediated by MGEs, which are able to move within or between DNA molecules (e.g., insertion sequences, transposons and gene cassettes/integrons), and genetic elements that are able to transfer between bacterial cells, such as plasmids and integrative conjugative elements [48]. In a recent review analysis of NDM-positive strains recovered worldwide, *bla*_NDM_ has been reported to be carried on plasmids (*n* = 355) with a variety of 20 replicon types in the Enterobacterales, a finding that highlights multiple acquisitions of *bla*_NDM_ by various plasmids via horizontal transfer [14]. *bla*_NDM-1_ was located mostly on incompatibility group (Inc) replicon types IncA/C, ColE, IncFIA, IncFIB, IncF IncII, IncHI1, IncHI3, IncHIB, IncL/M, IncN2, IncP, IncR, IncT, IncX1, IncX3, and IncY. In *K. pneumoniae,* a comparative genomic analysis of carbapenemase-harbouring plasmids from a European survey has revealed three modes of acquisition and transmission of antibiotic resistance genes via plasmids, one plasmid/multiple lineages, multiple plasmids/multiple lineages, and multiple plasmids/one lineage [49]. In the present study, both studied strains belonged to closely related MLSTs and carried IncFII (pECL_A)-like carrying *bla*_TEM-1,_ IncFII(pKPX1)-type-carrying *bla*_NDM-1_ and IncR plasmid sequences. Strain EC-ML559 harboured a Col440I-like carrying the quinolone resistance gene *qnrB19*. Strain EC-ML621 also harboured other plasmidic sequences found in other Enterobacterales, such as IncL/M (pMU407.1) carrying *bla*_CTX-M-14_, IncFII(Yp), IncFIB(pB171) and IncN [48,49]. For example, the IncL/M (pMU407.1) and IncFII replicon types have also been associated with *bla*_OXA-48_ and *bla*_KPC,_ respectively. These findings highlight the potential of the dispersion of these replicon type plasmids, which often carry other antimicrobial resistance and virulence genes, among different species in Enterobacterales, including species of the genus *Enterobacter*.

## 5. Conclusions

In the present study, a comprehensive analysis of the WGS assemblies of two representative *bla*_NDM-1_-producing outbreak isolates from Greece has shown that these isolates possess components of the bacterial cell surface, such as the enterobacterial common antigen, the colanic antigen, LPS-O-antigen, flagella, fimbriae and pili, which play a role in the pathogenicity of Gram-negative bacteria. Moreover, the presence of genomic islands, harbouring resistance genes to several antimicrobial classes and virulence factors, such as secretion systems and toxin–antitoxin systems, was revealed. Differences in the plasmids carrying β-lactamase genes between the two strains revealed diverse modes of acquisition and ongoing evolution of these mobile elements. In conclusion, WGS analysis of ECC strains would facilitate the development of methods for accurate identification and molecular serotyping schemes. Furthermore, WGS and comparative genomics of clinical and environmental ECC would elucidate the evolution, epidemiology, prevention and treatment of infectious disease caused by these broad-host-range niche-associated species.

## Figures and Tables

**Figure 1 antibiotics-12-01549-f001:**
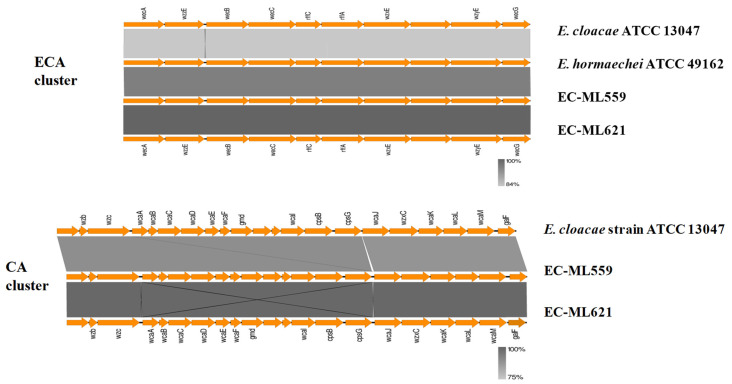
BlastN comparisons of the ECA and CA gene clusters of strains EC-ML559 and EC-ML621.

**Figure 2 antibiotics-12-01549-f002:**
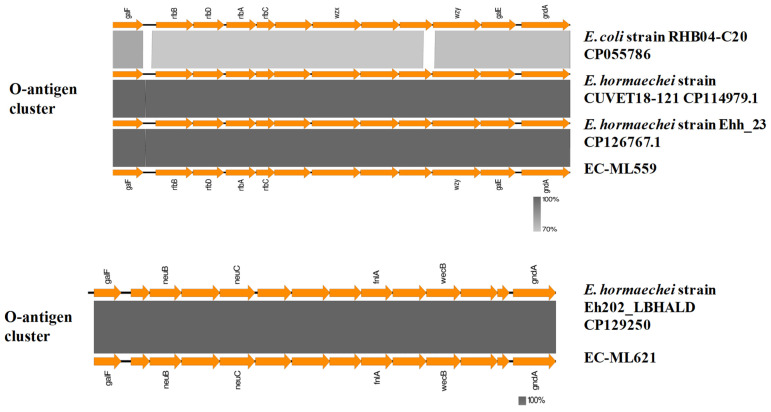
BlastN comparisons of the LPS O-antigen biosynthetic loci of strains EC-ML559 and EC-ML621.

**Figure 3 antibiotics-12-01549-f003:**
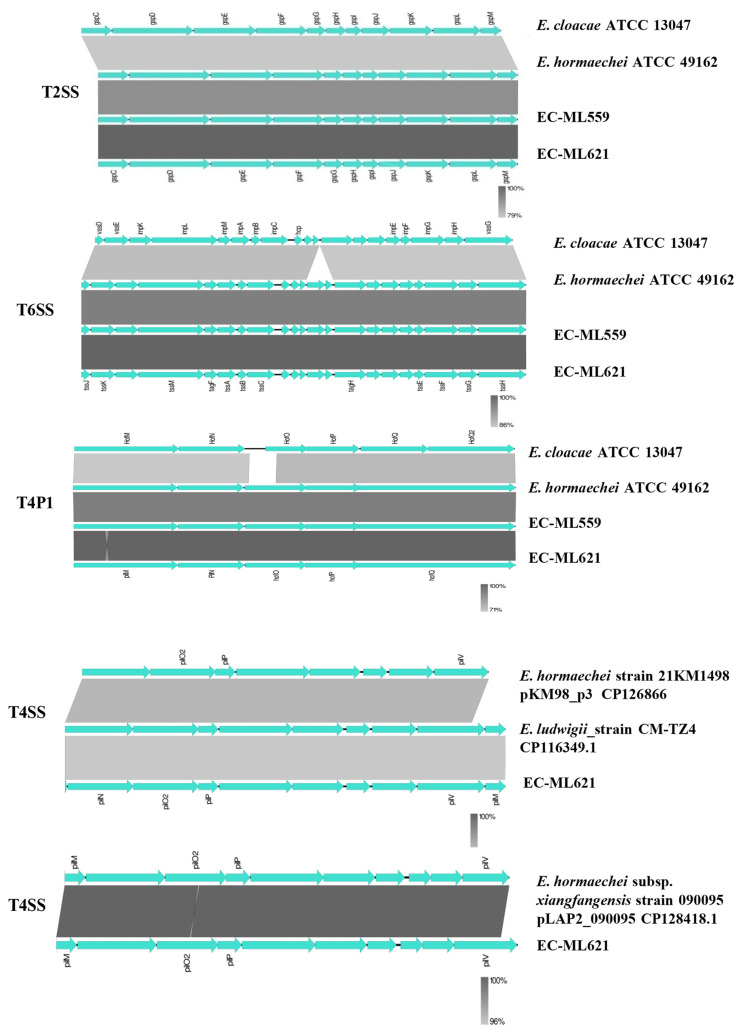
BlastN comparisons of T2SS, T4SS, T6SS and T4P1 secretion systems of strains EC-ML559 and EC-ML621.

**Table 1 antibiotics-12-01549-t001:** Characteristics of the WGS assemblies and MLST allelic profiles, antibiotic resistance and virulence genes in the predicted genomic islands of strains EC-ML559 and EC-ML621.

Characteristics #	EC-ML559	EC-ML621
WGS Assembly	Genbank Accession no. JARUPS000000000.1 Contigs: 74; total length: 4,958,007 bp, %GC: 55.07	Genbank Accession no. JASKGQ000000000 Contigs: 100, total length: 5,153,338 bp, %GC: 55.02
MLST ST (allelic profile)	ST182 (49-20-19-44-90-24-32)	**ST2143** (**554**-20-19-44-90-24-32)
Plasmids	IncFII(pECL_A), IncR **IncFII(pKPX1), Col440I**	IncFII(pECLA), IncR, **IncFII(Yp), IncL/M(pMU407), IncFIB(pB171), IncN**
ARGs	β-lactams: *bla*_ACT-16_, *bla*_NDM-1_, *bla*_TEM-1_, *bla*_OXA-1_ aminoglycosides: *aph(6)-Id, **aac(6′)-Ib3, aph(3′)-Ia,** aac(6′)-Ib-cr, aph(3″)Ib* chloramphenicol: *catB3* fosfomycin: *fosA* quaternary ammonium compounds: *qacE* quinolones: *OqxA, OqxB, **qnrB19*** trimethoprim: *dfrA14* sulfonamides: *sul1, **sul2*** macrolides: mrx(A), mphR(A), *mph(A)* rifampicin: ***ARR-3*** tetracyclines: ***tet(D)***	β-lactams: *bla*_ACT-16_, *bla*_NDM-1_, *bla*_TEM-1_, *bla*_OXA-1_, ***bla*_CTX-M-14_, *bla*_CTX-M-15_, *bla*_LAP-2_** aminoglycosides: *aph(6)-Id, **aac(3)-IId, aadA2,** aac(6′)-Ib-cr, aph(3″)-Ib* chloramphenicol: *catB3* fosfomycin: *fosA* quaternary ammonium compounds: *qacE* quinolones: *OqxA, OqxB, **qnrS1*** trimethoprim: ***dfrA12*** sulfonamides: *sul1* macrolides: mrx(A), mphR(A), *mph(A)*
HMRGs	Arsenic: *arsA, arsB, arsD, arsH, arsR,* Copper: *pcoA, pcoB, pcoC, pcoD, pcoE. pcoR, pcoS* Copper/Silver: *silA, silB, silC, silE, silP, silR, silS,* Tellurium: *terB, terC, terD, terW, **terZ***	Arsenic *arsA, arsB, arsD, arsH, arsR,* Copper: *pcoA, pcoB, pcoD, pcoC, pcoE, pcoR, pcoS,* Copper/Silver: *silA, silB, silC, silE, silP, silR, silS,* Tellurium: *terC, terD, terW* Mercury: ***merA, merC, merD, merE, merP, merR, merT***
TA-systems	*type II: **CcdA, CcdB,** HicA, HicB, ParD, RatA, RelB, RelE/ParE, VapB/VapC, Yaf; type IV: YeeU*	*type II: HicA, HicB, ParD, RatA, RelB, RelE/ParE,* *VapB/Vap; type IV: YeeU, **CbtA***
T1SS	*HlyD, TolC, MacA, MacB*	*HlyD, TolC, MacA, MacB*
T2SS	*gspC, gspD, gspE, gspF, gspG, gspH, gspI, gspJ,* *gspK, gspL, gspM* *secA, secB, secD, secE, secF, secG, secM, secY* *tatA, tatB, tatC, tatD, ftsY, yajC, yidC*	*gspC, gspD, gspE, gspF, gspG, gspH, gspI, gspJ, gspK, gspL, gspM* *secA, secB, secD, secE, secF, secG, secM, secY,* *tatA, tatB, tatC, tatD, ftsY, yajC, yidC*
T6SS	*tssH, tssG, tssF, tssE, tagJ, tagH, tssC, tssB, tssA, tagF, tssM, tssK, tssJ*	*tssH, tssG, tssF, tssE, tagJ, tagH, tssC, tssB, tssA, tagF, tssM, tssK, tssJ*
T4P1/T4SS	Locus 1: *hofC, hofB, HofQ, HofP, HofO, HofN/PilN, HofM/PilM*	Locus 1: *hofC, hofB, HofQ, HofP, HofO, HofN/PilN, HofM/PilM* Locus 2: ***pilN, pilO2, pilP, pilV, pilM*** Locus 3: ***pilM, pilN, pilO2, pilP***
Flagella	*fliZACDST*	*fliZACDST*
Fimbriae	P7E32_11190, P7E32_11195, P7E32_11200, P7E32_11205, P7E32_11210, P7E32_11215, P7E32_11220, P7E32_11225, P7E32_11230	P7F73_07975, P7F73_07980, P7F73_07985, P7F73_07990, P7F73_07995, P7F73_08000, P7F73_08005, P7F73_08010, P7F73_08015

# Differences among the strains are denoted in bold. ARGs: antimicrobial resistance genes; HMRGs: heavy-metal resistance genes.

**Table 2 antibiotics-12-01549-t002:** Identification of origin of transfers in DNA sequences of bacterial mobile genetic elements in strains EC-ML559 and EC-ML621.

Type	Contig GenBank Accession No.	Region/Gene (s)
Strain EC-ML559		
*oriT* region	JARUPS010000033	2799–2897
Relaxase	JARUPS010000032	*traI*
T4CP	JARUPS010000032	*traD*
T4SS—region 1	JARUPS010000031	*traN, trbC, traU, traW, traC, traV, traB, traK, traL, traE, traA*
T4SS—region 2	JARUPS010000023	*traM, traK, traJ, traH, traF, traB, traH, traG*
Strain EC-ML621		
*oriT* region	JASKGQ010000026	18926–18990
Relaxase	JASKGQ010000026	*traI*
T4CP	JASKGQ010000026	*traD*
T4SS—region 1	JASKGQ010000016	*tfc2, DotB,* *traH, traI, traJ, traK, traL, traM, traN, traO, traP, traQ, traR,* *traT, traU, traV, traW, traX, traY*
T4SS—region 2	JASKGQ010000022	*traH, traI, traJ, traK, traL, traM, traN, traO, traP, traQ, traR, traU,* *traW, traX, traY, trbC, trbB, trbA, trbN*
T4SS—region 3	JASKGQ010000024	*traM, traY, traA, traL, traE, traK, traB, traV, traC, traW, traU, * *trbC, traN, traF, traQ, traB, traH, traG, traD, traI, traX*

## Data Availability

The nucleotide sequences of the contigs of isolate EC-ML621 have been deposited in the Genome collection of *Enterobacter cloacae* PubMLST database, the Joint Genome Institute (JGI), Berkeley, USA, database under analysis Project GOLD ID Ga059773 and the National Center for Biotechnology Information (NCBI), National Institutes of Health, USA, database DDBJ/ENA/GenBank under the accession JASKGQ000000000. The version described in this paper is version JASKGQ010000000. The data presented in this study are available on request from the corresponding author.

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
