# Peer review of "Comprehensive Analysis of Virulence Determinants and Genomic Islands of blaNDM-1-Producing Enterobacter hormaechei Clinical Isolates from Greece"

_antibiotics, 2023, doi:10.3390/antibiotics12101549_

Round 1

Reviewer 1 Report

The manuscript is interesting and very well written. The methods and results are presented in a clear and orderly manner.

However I have some comments:

- The authors indicate in Table 1 that they detect IncL/M and FIIK plasmids in one of the Enterobacter. these plasmids are often found in other enterobacteria associated with other carbapenemases (e.g. IncL/M-OXA48 and IncFII-KPC). No comments are made in the discussion. It is important to highlight the role that these mobile genetic elements play in the dispersion of these resistance mechanisms among different species, including species of the genus Enterobacter.

- The authors could expand further on the discussion of the mechanisms of resistance to antibiotics (Carbapenemases, ESBLs and other beta-lactamases...). In the introduction this information is highlighted, but in the discussion mainly comments on virulence factors.

- I miss that antibiotic susceptibility data of the two strains studied are not included. A table could be added in the supplementary material with the results of the AST. 

Reviewer 2 Report

The manuscript entitled “Comprehensive analysis of virulence determinants and genomic islands of blaNDM-1-producing Enterobacter hormaechei clinical isolates from Greece” is well-described and it is of interest. In this study, a comprehensive analysis of the WGS assemblies of two blaNDM-1-producing ECC isolates that were involved in a clonal outbreak of blaNDM-1-producing ECC in a Greek teaching hospital, was carried out. This in-depth molecular study of these two isolates will help advance the knowledge regarding the evolution and epidemiology of this species and will help to better guide the prevention and treatment of the infectious disease caused by Enterobacter hormaechei.

Below, I pinpointed a few minor suggestions and corrections for improvement:

-        Lines 166, 221 and 223: ‘Similar to’ instead of “similar with”. The correct form is ‘similar to’. Check the whole manuscript to correct this grammar imprecision wherever it appears.

-        Lines 168-170: This sentence needs to be rephrased for legibility. Suggestion: The ECA cluster is found in many Gram-negative bacterial species and it encodes for proteins involved in many steps of ECA synthesis, such as the addition of ECA polysaccharide chains to a lipid carrier and the transportation of ECA to the bacterial cell surface.

-        Figures 1 and 2: Image quality must be improved. These figures are not legible.

-        Lines 273-276: Correct the grammar in this sentence. I suggest: A recent investigation of an outbreak in newborns with E. cloacae complex isolates revealed, by WGS, E bugandensis as the causative agent, and highlighted the need for better discrimination of Enterobacter species inside the ECC [48].

Minor editing of English language required.
